

# Determination of NO$_x$ emissions from Frankfurt Airport by optical spectroscopy (DOAS) – A feasibility study

Erna Frins[1], Reza Shaiganfar[2], Ulrich Platt[3], Thomas Wagner[2]

[1]Instituto de Física, Facultad de Ingeniería, Universidad de la República, Julio Herrera y Reissig 565, Montevideo, Uruguay

[2]Max-Planck Institut für Chemie, 55128 Mainz, Germany

[3]Institut für Umweltphysik, Universität Heidelberg, Im Neuenheimer Feld 229, 69120 Heidelberg, Germany

*Correspondence to*: Erna Frins (efrins@fing.edu.uy)

**Abstract.** Standard methods like in-situ measurements can hardly register NO$_x$ (=NO+NO$_2$) emissions from aircrafts during take-off, when engines run at high load and thus an important amount of fuel is consumed and most of the harmful emissions are produced . The goal of this work is to show that it is possible to measure aircraft emissions generated during take-off (and initial part of the climb) by a remote spectroscopic method like automobile - based - Differential Optical Absorption Spectroscopy (Mobile-DOAS), which uses scattered solar radiation in the blue spectral range (around 445 nm). In order to test its feasibility, total column measurements of NO$_2$ encircling Frankfurt Airport were carried out on 23 February 2012 using Mobile-DOAS. Also, NO$_x$ fluxes were derived from the NO$_2$ observations. Unlike standard mobile-DOAS measures using a spectrometer looking at zenith, the measurements were performed looking at 22° elevation angle leading to a roughly two to three times higher sensitivity compared to zenith observations. The origin of the observed NO$_2$ is discussed and the total NO$_x$ fluxes are calculated. As result of three round-trips encircling the Frankfurt Airport, the mean NO$_x$ flux was found to correlate with the number of aircrafts taking-off. Our results demonstrate that mobile-DOAS method is suitable for quantifying emissions from airports and to study their impact in the planetary boundary layer, which is most relevant concerning the impact on the environment and the human health.

25   ## 1   Introduction

Aircrafts, in contrast to other mobile sources, produce emissions that are released from ground level up to the free troposphere (and sometimes the lower stratosphere). An important amount of fuel is consumed during take-offs when engines run at nearly maximum load which leads to large NO$_x$ (= NO+NO$_2$) emissions (Herndon et al, 2004; Masiol and Harrison, 2014). The assessment of pollutants emissions from aircrafts is a complex issue, its analysis not only involves the exhaust emissions by landing and take-off, but also by queuing and taxiing from and to terminal gates. Also, other ground services associated with the transportation of passengers and airfreight, luggage transport from and to terminal gates as well as the transportation for refuelling aircrafts and catering service produce pollutant emissions at airports which can interfere with measurements performed using in-situ instruments. Nowadays, some of these services use electric transport, thereby reducing emissions at the airport. However this improvement is not widespread and does not apply at all airports.

35   Studies of trace gas emissions from aircrafts mainly relied on inventories (see e.g. Winther et al., 2015 and references therein) and air quality data obtained from fixed observation sites, but little observation data under real conditions are available. Pison





et al. (2004) analysed the impact of aircraft emissions on photo-oxidant by means of the chemistry transport model CHIMERE (http://www.lmd.polytechnique.fr/chimere/) over the Paris area. Using a three-dimensional aircraft emission inventory, they compare ozone surface concentrations obtained with and without these emissions based on data obtained from three sites: one in Paris South-East and from two rural sites North and South of Paris. They conclude that $NO_x$ emissions from air traffic have a more important impact than VOC emissions.

Other assessments of airport emissions have focused on emission factors and fuel consumption (see for instance the review by Kurniawan and Khardi, 2011, and Masiol and Harrison, 2014). The more common methodology is to estimate pollutants emissions considering the landing and take-off cycle (LTO), defined by the International Civil Aviation Organization (ICAO) (EPA, 2012). Clearly the lack of information of the type of fuel and actual flight movement data (taxi, queuing time, aging effects, etc) are some of the main limitations of these studies. Indeed, knowing the technical details of the aircraft is of limited value as long as the operation of the airport is unknown. Herndon et al. (2004) performed ground based in-situ measurements of NO and $NO_2$ by means of a Tunable Diode Laser Differential Absorption Spectroscopy (TILDAS) and of $CO_2$ using a commercial non-dispersive infrared absorption technique. They compared the ICAO reference Emission Indices (EI) with EI derived from their campaign measured during taxi and take-off operation of three aircraft. The ICAO quotient of emissions indices between take-off and taxi is of the order of 6, while Herndon et al. (2004, Table 1) obtained a quotient in an interval of the order of 7-18. This shows that the emissions during take-offs are considerable larger than during taxi, and also, that the measurement of emissions from aircraft is a complex task fraught with uncertainty.

Detecting and quantifying $NO_x$ emissions from an aircraft using in-situ instruments located outside the facilities needs special attention because of the presence of other $NO_x$ sources in the airport and close to it (e.g. emission from motorways). The main difficulty is to detect and extract the contribution made by the airport from other local $NO_x$ sources. Carslaw et al. (2006) presented an approach to overcome this problem based on a network of seven measuring sites close to an airport. On one side they proceed graphically (polar plots) generating a sort of pollution rose, which helps to discriminate different sources. On the other hand they correlate these plots with the airport activity, which enables data filtering techniques to verify the presence of aircraft sources. They study the emissions from the Heathrow Airport and claim that $NO_x$ emitted by aircrafts can be detected to at least 2.6 km from the airport.

Measurement campaigns by remote sensing were performed by Schäfer et al. (2003) on idling operation of aircrafts at major European airports to assess emission indices. During idle operation they observed NO at the engine exit with a FTIR spectrometer and $NO_2$ some 50 m from the engine nozzle with a Long-path-DOAS system.

All the above mentioned issues largely stem from the fact that ground measurements do not measure the total emissions (including those at higher altitude), but only the ground-level concentrations. At the near ground atmosphere, the proportion of aircraft emissions might be rather low comparing with all the emissions due to the airport activity and surroundings. These problems can be overcome with Mobile-DOAS measurements. Mobile-DOAS is an established method to quantify emissions from industrial facilities and urban areas (Galle et al. 2002; McGonigle et al. 2003; Rivera et al., 2009; Johansson et al. 2008; Ibrahim et al. 2010, Wang et al., 2012, Frins et al., 2014).

In this report we present the results of $NO_2$ measurements encircling the Frankfurt Airport using Multi Axis - DOAS (MAX-DOAS) from an automobile platform. In the next Section we describe the method to retrieve the $NO_x$ emissions. The experimental results and possible error sources are presented in Section 3. Conclusions are presented in Section 4





## 2    Method

### 2.1    Mobile-DOAS data acquisition

As mentioned above mobile-DOAS has been used by several investigations to assess emissions from anthropogenic sources. MAX-DOAS observations at fixed locations are usually performed at low elevation angles (close to the horizon) to improve
the sensitivity of the method. In contrast, for mobile-MAX-DOAS measurements, observations at low angles are problematic, because in many cases the open sky is blocked by buildings or trees. Thus usually higher elevation angles (typically between 20° and 30°) are used (Wagner et al. (2010)). From these observations the tropospheric vertical column density ($V$, i.e. the vertically integrated concentration) of trace gases can be determined.

The present study is based on measurements performed on 23 February 2012 along the route encircling the Frankfurt
International Airport, Germany (50° N, 8.5° E, see map in Fig. 1). A fully automated MAX-DOAS instrument (Hoffmann Messtechnik GmbH) was mounted on the top of a car, which was driven along two main motorways and highways to encircle the complete airport area (except the new runway in the North, which is only used for landing). The instrument was controlled by a script running under the software DOASIS (Kraus, 2006).

The MAX-DOAS instrument consists of a spectrometer (USB2000+, Ocean Optics Inc.) with temperature control and a
stepper motor to guide the direction of observation. A quartz lens of 40 mm focal length coupled scattered light into the spectrometer through a quartz fibre bundle of four optical fibres of 200 µm diameter each. The resulting aperture angle of the telescope was ~ 1°. The spectrometer was designed for a spectral range from about 315 nm to 460 nm and spectral resolution of ~ 0.5 nm. In order to avoid instabilities during acquisition (and to reduce the detector dark signal) the spectrometer was thermostated to 2°C during all measurements. The telescope viewing direction was in the plane spanned by the vertical and
the direction of travel. Spectral data was accompanied with GPS data to give the temporal and spatial information for each measurement point on the path around the airport. Spectra were sequentially acquired ten times at 22° elevation angle alternating with measurements at 90° and 45°. The duration of an individual measurements was about 10-25 s resulting from the addition of about 150-500 individual scans.

### 2.2    Vertical Column Density Retrieval

The measured spectra were analyzed using the DOAS-method (Platt and Stutz, 2008) implemented in the WINDOAS-tool developed at the BIRA-IASB (Fayt and Roozendael, 2001). The spectra were analyzed using a Fraunhofer reference spectrum in an outlying position of the plume. Thus, as result of the analysis a differential slant column density (dSCD) is obtained. To analyse $NO_2$ a fitting window between 430-460 nm was chosen, where strong structured absorption features of $NO_2$ are observed. Additionally, the cross sections of $O_4$ (Greenblatt et al., 1990) and $NO_2$ (Vandaele et al., 1998) at 243 K,
$O_3$ (Bogumil et al., 2003) at 241 K and water vapour at 290 K (Rothman et al., 2005) were used. A synthetic Ring spectrum calculated using the DOASIS software was also included in the spectra evaluation (Kraus, 2006; Wagner et al., 2009) and a third degree polynomial was fitted. An example of the spectral retrieval is shown in Fig. 2.

The relevant magnitude to estimate the flux of a source is the vertical column density ($V$) of the tropospheric component which is related with the measured slant column density ($S$) through the so called air mass factor ($A$) (Noxon et al., 1979;
Solomon et al., 1987; Marquard et al., 2000). In this study, slant column densities were acquired at elevation angles $\alpha = 22°$ and then converted to vertical column densities considering a geometrical factor:





$$V_{trop} = \frac{S_{trop}(\alpha)}{A_{trop}(\alpha)} \, . \tag{3}$$

Here $V_{trop}$ indicates the tropospheric vertical column density and $S_{trop}$ the tropospheric slant column density. The $S_{trop}$ is derived according to the method described in Wagner et al. (2010). The tropospheric air mass factor is determined by the so called geometric approximation:

$$A_{trop} \approx \frac{1}{\sin \alpha} \, . \tag{4}$$

For an elevation angle of 22° a tropospheric air mass factor of 2.67 is obtained.

### 2.3    Flux calculation

From the vertical tropospheric trace gas columns derived from mobile-MAX-DOAS observations of encircled emission
sources and the wind fields the flux $F$ of the trace gas can be calculated as follows:

$$F = \int_C VCD(x, y)_{trop} \, \vec{v} \cdot \hat{n} \, dl \, , \tag{5}$$

where $\vec{v}$ is the average wind vector and $\hat{n} \, dl$ is the surface element vector orthogonal to the driving direction (Johansson et al., 2008, Rivera et al., 2009, Ibrahim et al. 2010, Frins et al. 2014). The basic assumption is that the trace gases move at the same speed and direction as the air masses. In our study the wind speed and direction along the area of interest is obtained
from the meteorological stations of the Frankfurt Airport. During the time of the measurements the wind speed was on average 4.5 m s$^{-1}$, and the wind direction was 200°. Since part of the detected emissions occur after the take-off (see Section 3.2), we calculate average wind speed and direction according to the typical change of wind speed and direction with altitude (Ekman spiral). Thus for the estimation of the emission flux (eq. 5), we use an average wind speed of 6.7 m s$^{-1}$ and wind direction of 207.5°.


### 2.4    Conversion of NO to NO$_2$

At the nozzle exit of an aircraft nitric oxide (NO) is the main nitrogen containing oxide. However only nitrogen dioxide (NO$_2$) presents strong absorption features in the UV and visible spectral range (> 300 nm) where solar radiation is available at Earth's surface, thus, the NO$_2$-dSCD can be easily extracted from sunlight spectra obtained from MAX-DOAS observations.
This is not the case for the trace gas NO with UV transitions in the 200-230 nm wavelength range. However, the major fraction of NO is rather rapidly (time constant of the order of 1 minute) oxidized to NO$_2$ by atmospheric ozone (Seinfeld and Pandis, 2006). We assume that the strong turbulences produced (by landings but mainly) by take offs boost the mixing of the emitted gases with atmospheric air. Thus, the emitted NO$_x$ fluxes are estimated by assuming that the measured NO$_2$ originates mainly from nitric oxide and that both trace gases are transported at the speed of the air masses to the perimeter of the
measurement route (at 2-6 km away from the runways). Depending on the ozone concentration and NO$_2$ photolysis rate the fraction of both species, NO and NO$_2$, will be affected in the atmosphere and observed during the circle around the airport.





To estimate the total $NO_x$ emitted at the runways the Leighton-ratio L (= [NO]/[NO$_2$]) is calculated using the mixing ratios provided by the monitors located at the airport. The corresponding values for the three loops are presented in Table 1. For the determination of the $NO_x$ emissions from the measured $NO_2$ fluxes (eq. 2) the individual Leighton ratios during the three loops were used.

**2.5    Effect of finite lifetime of $NO_x$**

The lifetime of $NO_x$ depends on meteorological parameters like temperature and radiation. Since the measurements were performed in winter we assume a rather long $NO_x$ lifetime of 20 hours (Beirle et al., 2003; Seinfeld and Pandis, 2006). Taking into account the rather short transport times of the air masses from the locations of the emission to the measurement we conclude that the effect of $NO_x$ destruction can be neglected for our measurements.

**3    Experimental results**

**3.1    $NO_2$ tropospheric Vertical Column Density and NOx fluxes**

The airport was three times surrounded driving along the highways (Fig. 1). The emitted plume has its origin mainly at the two runways arranged in the East-West direction called Runways North and South (also called runways 25C and 25L,

respectively) used for landings and take offs and the Runway West (also called runway 18) used for take offs in southbound direction only.

The exposure time of spectra was between 15 s to 25 s covering regularly most of the route. The first roundtrip took place in the interval 13:15-13:44 UTC, the second roundtrip in the interval 13:45-14:21 UTC, and the third one in the interval 14:22-14:44 UTC.

The resulting vertical column density of $NO_2$ derived from the MAX-DOAS observations along the route encircling the airport are represented in Figure 3 by coloured dots. According with the meteorological station located at the Frankfurt Airport, the prevailing wind direction was SSW (200°) and constant throughout the time it took to perform the measurements. Thus, the highest $NO_2$ vertical column densities are observed on the northern part of the route (NNE).

The observed $NO_2$ flux was calculated as described in Section 2.3 and the results are summarized in Table 1.

**3.2    Uncertainties by the flux calculation**

According to Eq. 5 several uncertainties directly affect the derived emissions. The uncertainty of the tropospheric $NO_2$ vertical column density is typically <20% (Shaiganfar et al., 2011). For the wind data we used observations from the meteorological weather stations at the airport. The wind direction was stable during the three circles around the airport, which is evident from the observed vertical column densities represented in Figure 3. The variation of the wind speed during the

three circles was about 25%. Thus uncertainty related to the variability of the wind field is estimated (based on eq. 5) to about 30%. Further errors are caused by the missing measuring points along the route especially the third circle (14:22-14:49), when driven along the eastern side of the airport. We estimated the corresponding error to about 16% for the first and second circle, and about 24% for the third circle based on the method presented in Shaiganfar et al. (2016).



While the effect of the limited $NO_x$ lifetime can be neglected because of the short transportation time between the locations of the emission and measurement, the uncertainty of the partitioning between NO and $NO_2$ affects the derived emissions. We estimate this uncertainty to 15% (Shaiganfar et al., 2016).

Assuming that the different error sources are independent from each other we derive total uncertainties of the derived $NO_x$ emissions between 40% and 45%. Despite this relatively large uncertainty the proposed method provides information about airmasses at an otherwise essentially inaccessible altitude range.

### 3.3    Considerations about the observed emissions from aircrafts during take-off

As mentioned above our assumption is that the main emission of $NO_x$ comes from the take-off of the aircrafts. During this phase of the flight, the speed of an aircraft changes from zero to around 280 km h$^{-1}$ (77 m s$^{-1}$). This means that after moving ~ 2 km along the runway the aircraft spent ~ 52 s on the ground before lift-off (see Fig. 4). The take off operation is a high power operation and depends on several factors such as type of aircraft, load and weather conditions. To roughly estimate how long the plane stays within the area encircled by the mobile-DOAS platform, we simplify the description of the take-off operation after lift-off considering an elevation angle of 15°. The distance between the lift-off position (rotation of the

aircraft) and the motorway used by the car is approximately on average 4.3 km. This means that the aircraft climbs during ~1 min before leaving the encircled area. Under these assumptions the airplane reaches an altitude of approximately 1150 m and was ~ 110 s (including the acceleration phase on the runway) expelling gases under these extreme conditions.

Typically MAX-DOAS observations are sensitive up to an altitude of about 3 km (Frieß et al., 2006; Shaiganfar et al., 2015). Since the observation angle of the instrument is 22° with respect to the ground (looking backward the travel direction), we

can ensure that the $NO_x$ emissions within the encircled area are in the field of view of the mobile-DOAS instrument.

During the field test around the airport the speed of the air masses (i.e., the wind speed) was 4.5 m s$^{-1}$ from the SSW-direction. Thus, in order to establish a correlation between the number of taking-offs and the $NO_x$-emission, the Runway West (i.e. runway 18) requires special consideration since the air masses travelled a distance between 2 - 6 km (the aircraft moves along the runway changing its distance to the highway) from emission to the measurement site at the north portion of

the driving route of the mobile-DOAS, which implies a time delay of the order between 10-30 min.

Thus for example, during the first roundtrip encircling the airport the car reaches the north portion of the circle approximately at 13:30 UTC. Then, in order to establish a correspondence between the measured $NO_x$-flux and the number of take-offs one has to consider the take-offs from the Runway West in a time interval of 10-30 min previous to 13:30 UTC, while the air masses coming from the take-offs from Runways North basically have not time delay (or it is quite small). The same

considerations have to be taken into account for the second and third roundtrip.

Table 2 shows the $NO_x$-fluxes taken from Table 1 and the number of take-offs from the different runways (taking into account the time delay discussed above).

A good correlation ($r^2$ = 0.75) between the total take-offs number and the $NO_x$-fluxes measured with mobile – DOAS is found. The large estimate error in the number of take-offs is due to the fact that the information about take-offs is available at

15 min intervals. From the linear regression of the $NO_x$ emissions versus the number of take-offs we derive a slope of about $8 \times 10^{23}$ molecules s$^{-1}$ per take-off.





## 4       Discussion and conclusions

Besides CO$_2$ and CO, NO$_X$ are also relevant exhaust emissions of aircrafts (IPCC http://www.ipcc-nggip.iges.or.jp/public/gp/bgp/2_5_Aircraft.pdf). NO$_x$ are primarily produced mainly as nitric oxide and rapidly oxidized to

NO$_2$. During the start-up phase of an aircraft (i.e, take-off and climbing at an altitude ~ 1150m) a significant amount of NO$_x$ is emitted. Mobile-DOAS is well suited for its detection and quantification, because it is also sensitive for the emissions at higher altitudes (but still within the boundary layer) while measurements performed at ground level with *in situ* instruments or remote methods from a fix location, can only partially observe the take-off operation. Indeed, emissions in the planetary boundary layer (PBL, i.e. from ground level to about 2.0 km height) like take-offs events are relevant for the air pollution at

ground-level. Another important advantage of this method is that it allows the detection of gases from a remote location (some kilometres away from the source), which it is not of small relevance since airports working areas are particularly sensitive to any intrusion of people and instruments within the facility.

In our study we have performed three round-trips to estimate NO$_x$ emissions, and we found values between 4.6 x10$^{24}$ and 9.1 x10$^{24}$ molecules s$^{-1}$. From the regression analysis of the NO$_x$ emissions versus the number of take-offs we derive a correlation

coefficient (r²) of 0.75. From the slope we derive a mean NO$_x$ emission of (8 ± 4) x10$^{23}$ molecules s$^{-1}$ per take-off. However, this value should be treated with care, particularly due to the lack of information about the type of the aircrafts taking off during these periods of time. Future studies should also try to estimate the contribution of the NO$_x$ emissions from taxi and landing. However, descent operation could be considered as an "idle" operation until reaching approximately 300 to 500 meters above ground on the approach to the landing runway. Thus, from a fuel consumption perspective it is a low power

event and low emissions.

In conclusion, we have demonstrated that the mobile-DOAS method is suitable for quantifying airport NO$_x$-emissions generated during take-offs and part of the climbing phase i.e. the emissions that take place close to the ground and up to the free troposphere.

**Acknowledgements**

This work was partially supported by Comisión Sectorial de Investigación Científica (CSIC), Universidad de la República (Uruguay), PEDECIBA (Programa de Apoyo a las Ciencias Básicas). E.F. acknowledges funding by the L'Oreal National Award for Women in Science. The authors acknowledge Frankfurt Airport for providing air quality data from its measuring stations. We are also very thankful to Manfred Maiß for providing the runway usage data, and Markus Sommerfeld and

Klaus-Peter Heue for air quality data.

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

20

25





**Table 1.** Summery of the total $NO_2$ and $NO_x$ fluxes for the different roundtrips

| Universal time | $NO_2$-Flux [molecules s$^{-1}$] | Uncertainty [molec. s$^{-1}$] | Leighton ratio | $NO_x$-Flux [molecules s$^{-1}$] | Uncertainty [molec. s$^{-1}$] |
|---|---|---|---|---|---|
| 13:15-13:44 | $6.7 \times 10^{24}$ (0.51 kg s$^{-1}$) | $2.7 \times 10^{24}$ (0.20 kg s$^{-1}$) | ~ 0.34 | $9.1 \times 10^{24}$ (0.69 kg s$^{-1}$) | $3.6 \times 10^{24}$ 5 (0.28 kg s$^{-1}$) |
| 13:45-14:21 | $3.5 \times 10^{24}$ (0.27 kg s$^{-1}$) | $1.4) \times 10^{24}$ (0.11 kg s$^{-1}$) | ~ 0.30 | $4.6 \times 10^{24}$ (0.35 kg s$^{-1}$) | $1.8 \times 10^{24}$ (0.14 kg s$^{-1}$) |
| 14:22-14:49 | $6.6 \times 10^{24}$ (0.50 kg s$^{-1}$) | 3.0 (0.23 kg s$^{-1}$) | ~ 0.28 | $8.5 \times 10^{24}$ (0.65 kg s$^{-1}$) | $3.8 \times 10^{24}$ (0.29 kg s$^{-1}$) 10 |

.



**Table 2.** Number of take-offs from the different runways

| Roundtrip | NO$_x$-Flux [molecules s$^{-1}$] | Take-offs Runway West | Take-offs Runway North | Total Take-offs (error: ±2) |
|---|---|---|---|---|
| 1 | $(9.1 \pm 3.6) \times 10^{24}$ | 5 | 5 | 10 |
| 2 | $(4.6 \pm 1.8) \times 10^{24}$ | 5 | 0 | 5 |
| 3 | $(8.5 \pm 3.8) \times 10^{24}$ | 6 | 1 | 7 |





**Figure 1**

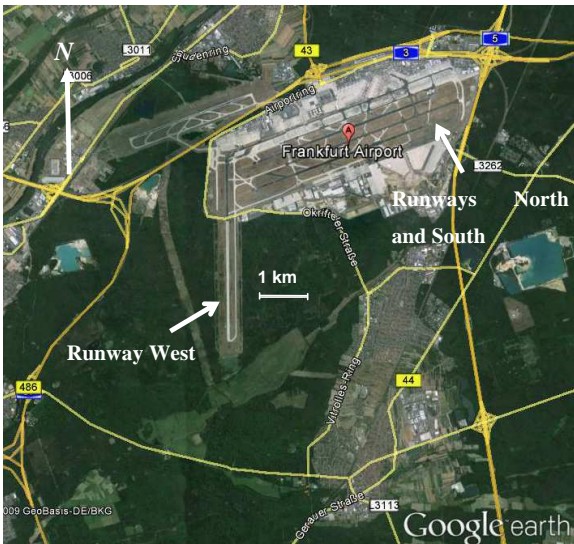

**Figure 1:** Arrival and departure runways of the Frankfurt Airport within the motorways driven during
the measurements: Runway West (take off only), Runways North and South.





**Figure 2**

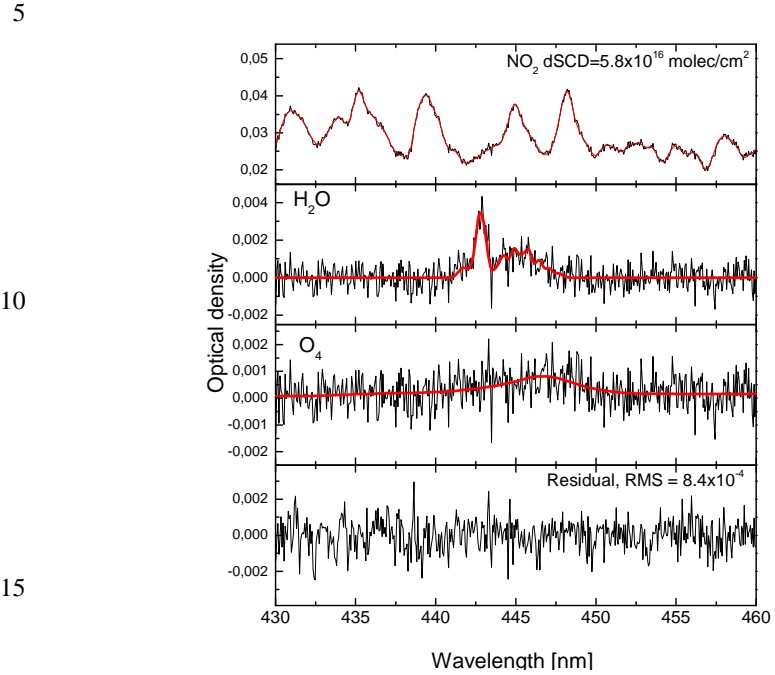

**Figure 2:** An example of DOAS fit for NO$_2$ retrieval. The spectrum was recorded on 23 February 2012

at 13:39 UT and at 22° elevation angle. The red lines are the molecular absorption cross sections scaled

20   to the detected absorptions in the measured spectrum. The black lines are the residual structure added to

the retrieved absorption structure.

25





5    **Figure 3 (a)**

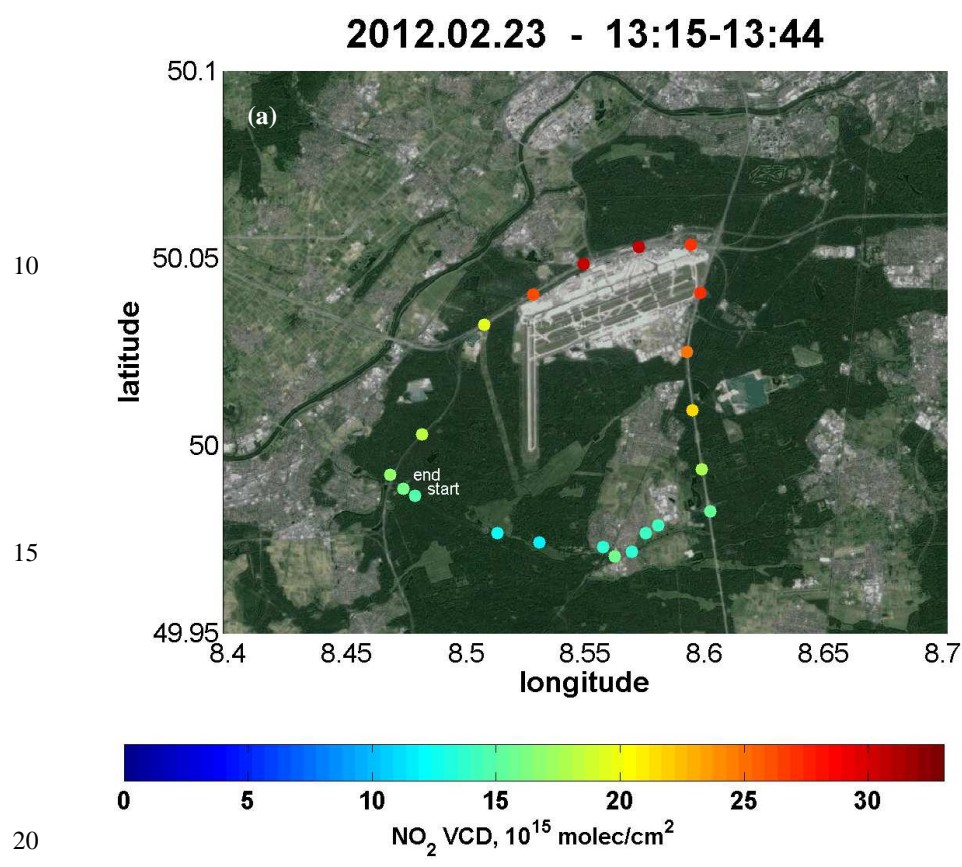



5 **Figure 3 (b)**

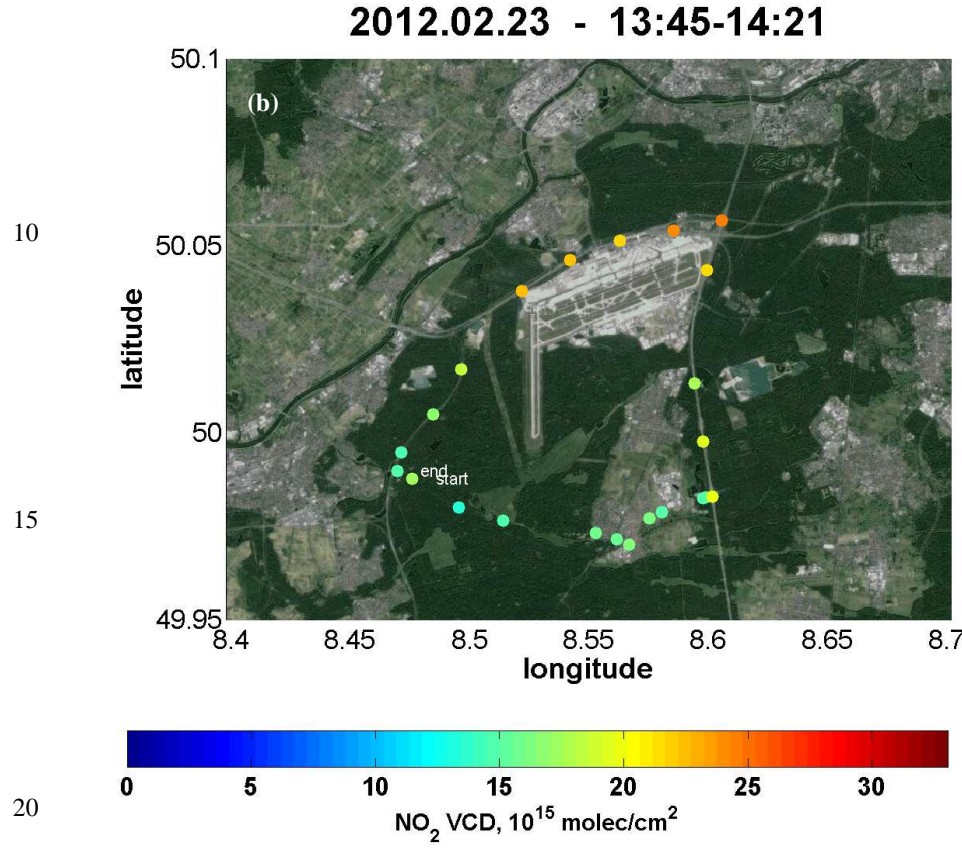





5    **Figure 3 (c)**

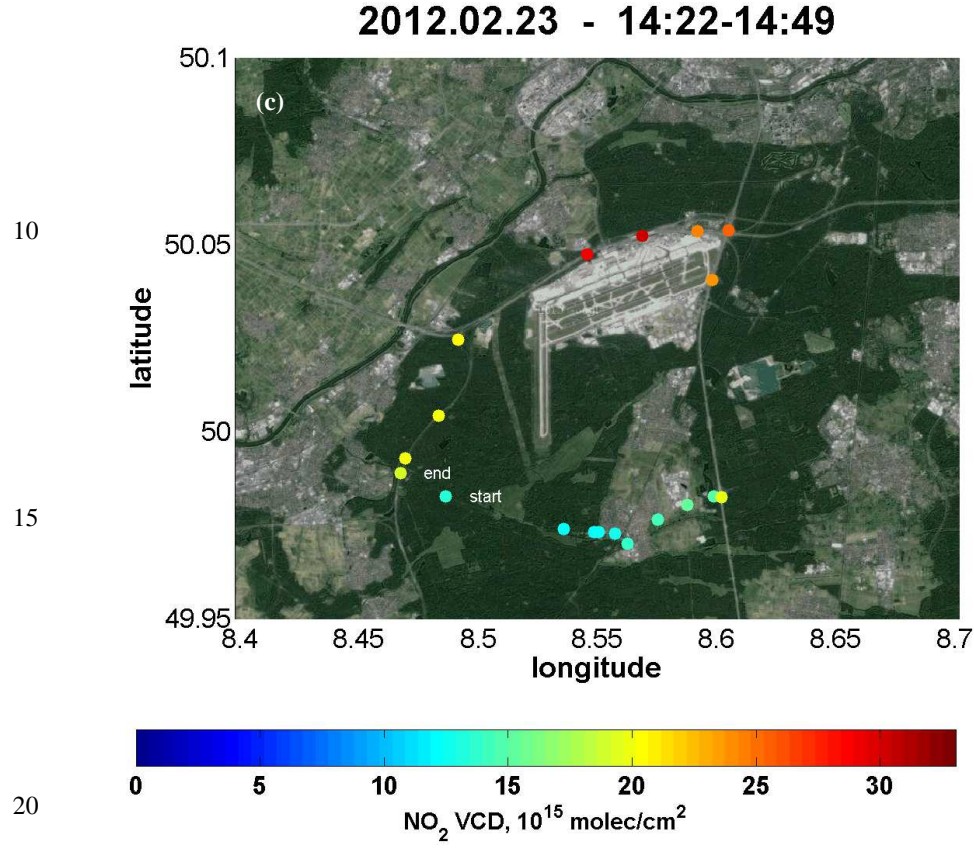

**Figure 3:** Tropospheric vertical column densities of $NO_2$ derived from the mobile-DOAS observations

25    along the route encircling the airport.





**Figure 4**

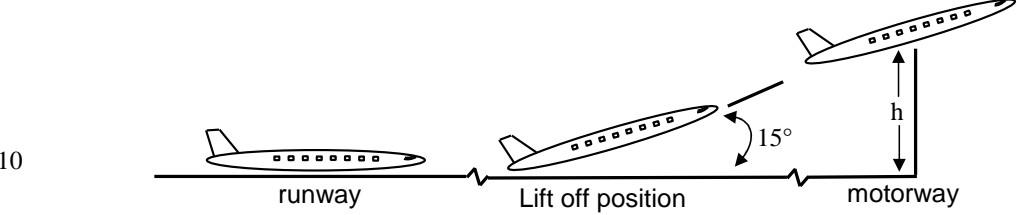

**Figure 4:** Our approach to describe the take off operation of a Boing 747. After a ~ 2 km runway it reaches the speed of ~280 km h$^{-1}$ and passes over the motorway at an altitude of approximately 1150 m.

