# Peer review of "Determination of $NO_x$ emissions from Frankfurt Airport by optical spectroscopy (DOAS) – A feasibility study"

_Atmospheric Measurement Techniques, 2016_

## Referee Comment (RC1) · Anonymous Referee #1 · 10 Aug 2016

This paper is slightly improved from the earlier rejected version on the same topic. The previous version was rejected for many good reasons and the authors have decided to consider only some of these. This paper is a long way from being publishable and is currently unacceptable for AMT.

The addition of the spectral fits is a major addition to this new manuscript. The co-adding of spectra is a key feature of the method considering the shallow well depth of the USB2000+ spectrometer.

The paper remain short on details and important error sources are neglected or not

properly considered.

The air mass factor (AMF) is not valid and no associated uncertainty is considered.

The authors are misinterpreting/overstating the significance of the correlation between NO2 flux and number of airplane take-offs.

Some important revisions are suggested below.

General comments

Since only one elevation angle is used, the use of the term MAXDOAS is not appropriate (e.g. p2L35). 'Off-axis DOAS' would be appropriate.

The air mass factor used is too simple since it only depends on the viewing elevation angle. The authors should demonstrate that the SZA has no effect on the air mass factor. They could consider SZAs between 0 and 90 deg. I am quite certain they will find some importance. The solar elevation angle range (in degrees) for the three measurement periods in chronological order was: 28.5-29.5, 26.2-28.5, and 24-26. The near-equality of the viewing and solar elevation angles appears to be a fluke that could be noted to possibly defend their current AMF equation (Eq. 2).

Section 2.1 should discuss the cloud conditions. This is relevant for the Ring effect and the pathlength for NO2 absorption, particularly since the authors assume a geometric air mass factor based on the viewing elevation angle. Did there appear to be cloud anywhere on this day? Was there ever a cloud in the FOV on this afternoon? The second question is most important, especially if the cloud is below the top of the boundary layer. Was there snow on the ground? The daytime maximum temperature was 7 deg C on Feb 23rd, 2012 at Frankfurt airport.

Section 3.1 is where Figure 2 should be called, not in Sect. 2. One sentence should be provided to give the typical relative uncertainty of the ∼60 NO2 SCDs corresponding to the same number of coloured dots in the panels of Fig. 3.

Correlation: three points is a bare minimum for using the correlation statistic. The required correlation coefficient to be statistically significant using an alpha of 0.05 is 0.921 for a one-tailed test. The r for the correlation between NO2 flux and number of airplane take-offs is 0.87. Thus no discussion of correlation is appropriate in my opinion.

I also think the experiment was not done correctly. The viewing azimuth angle should be controlled relative to the airport, not relative to the car/road direction. This is a source of error to be discussed. I believe a zenith viewing spectrometer would have been better: while less sensitive, it is more suited to the derivation of fluxes using winds (Eq. 5). The low elevation angle means the instrument is remotely sensing a relatively large volume rather than the local VCD at each dot in Fig. 3. Whether viewing is to the zenith or at a low elevation angle, the large SZA means that a large distance is sampled along the incoming path. I don't understand why the n*dl is defined as being for the direction orthogonal to the driving direction. What is the typical n*dl? Is it 2-6 km (p6L23)? If so, this should be first stated in sect 2.3.

The method also assumes that there is no dispersion of the aircraft NOx from the "nozzle exit" to the car-borne sensor. For example as the car heads north to the northeast corner of the circuit, dispersion to the west or vertically would lead to a bias. This assumption could be checked with a dispersion model. If dispersion is an issue and given the strong turbulence (p4L25) it might be, the VCDs at the circumference of the circle may not reflect the VCDs of the runway (i.e. inside the circle) and this would be a source of error in applying Eq. 5.

The description of the flux calculation in sect 2.3 is severely lacking. The upwind measurements tell you nothing about the airport emission. The authors should use Figure 3 to point out which measurements are considered upwind and which are downwind for the flux estimates. Do the authors use the upwind measurements to correct for the NOx flux from local non-airport sources? This seems like a good idea.

[Figure]

In spite of my comment on the previous rejected manuscript, the authors continue to refer to a study in Delhi for the tropospheric NO2 VCD uncertainty. This is absurd.

Specific science comments

p3L27 "outlying position of the plume" should be reworded. Are the authors trying to communicate that the reference spectrum was acquired in a location not influenced by NOx emissions from the airport? More details about the reference spectrum should be provided. For example, was it obtained at the same elevation angle? Was it obtained on the same day? Both are relevant to its Ring effect signature. Figure 2 should show the fitted Ring differential optical depth (DOD) as well for the same co-added spectrum. The y-axis on Figure 2 should be "Differential optical density", not "optical density".

Figure 3 shows that the NO2 VCDs are always positive. The authors could indicate this in the manuscript to support the point that the reference spectrum has a weaker NO2 absorption signal. I wonder if the authors used a zenith-sky spectrum as the reference spectrum. As noted by the authors, this would reduce the sensitivity to local sources surrounding the airport. The amplitude of the NO2 absorption signal in the reference spectrum is another source of bias that is completely overlooked.

Figure 3 - Was the overhead image taken by the authors? If not, the source should be credited. If the date that the image was taken is known, it should be provided.

p4L18 The Ekman spiral is interesting but the authors need to provide more details (or a reference on how the average wind speed and direction were calculated). I certainly could not repeat their calculation given the lack of details. Over which altitude range is the averaging done?

p5L3 According to http://www.atmospheric-measurement-techniques.net/for_authors/manuscript_preparation.html "Works cited in a manuscript should be accepted for publication or published already." Thus, the important reference to Shaiganfar et al., 2016 is not appropriate. In fact, 2016 is unlikely to be the year of

publication if the manuscript were submitted and accepted.

Minor comments

p1L14 "automobile - based" -> "automobile-based"

p1L17 "standard mobile-DOAS" -> "standard mobile-DOAS that"

p1L29 'pollutant' is not an adjective but is used as one. "pollutant emissions"-> "emissions of pollutants"

p1L32 "pollutant emissions" -> "polluting emissions"

p2L22 (and elswhere in the manuscript) Leading prepositional phrases such as "On one side" should be followed by a comma, a rule the authors have correctly followed on p7L21

p4L25 The sentence starting with "However, the major" should be the second sentence of the paragraph. Then "However, only" could be deleted from the next sentence leaving it starting with "NO2 presents ..." The chemical formulae NO and NO2 in parentheses together with the chemical names nitric oxide and nitrogen dioxide should be provided in the introduction.

p5L25 "Uncertainties by"->"Uncertainties in"

p6L17 Move "∼110 s" after "expelling gases".

p6L29 not -> no

p7L6 for the -> to the

p7L33 The '2' in 'NO2' should be subscripted here and probably elsewhere.

p8L23 The link to Kraus's thesis is broken or incorrect.
* * *

---

## Referee Comment (RC2) · Anonymous Referee #2 · 26 Aug 2016

General comments

The paper is in general well written and provides new insights regarding the application of optical remote sensing, specifically the DOAS technique, to quantify emissions from an airport. Determining the pollution burden of airports pose a challenge due to the variability of emissions, either due to timing of taxiing and take off as well as differences inherent to types of aircrafts and airport operation. The contribution of this paper is valuable since it provides a clear example of the application of a measurement technique that can help to overcome these challenges.

Specific comments

[Figure]

Please comment about the advantage of using an off-axis measurement geometry instead of a zenith-viewing geometry.

Please provide more information regarding the used Fraunhofer reference. When and in which elevation angle was taken?

Please comment about the validity of using a geometrical factor to convert from SCDs to VCDs.

For correlation analysis purposes it would have been advisable to have more measurement points, however it is comprehensible that this was a feasibility study and few measurements were conducted in order to demonstrate the measurement methodology.

Technical comments

Page 11. "Summary" instead of "Summery" Page 18. "Boeing" instead of "Boing"
* * *

---

## Referee Comment (RC3) · Anonymous Referee #3 · 23 Sep 2016

In their manuscript, the authors report on car DOAS measurements of NO2 around Frankfurt airport. Using the vertical columns derived from their observations in combination with wind data they estimate the NOx emissions of the airport during three rounds. With additional assumptions on aircraft movements and vertical distribution, they then derive rough estimates of emissions per take-off.

The topic of the paper is interesting as aircraft emissions of NOx are poorly constrained and car DOAS measurements could be a simple way of adding information on this topic, in particular as they integrate vertically. The manuscript is clearly written and fits into the scope of AMT. However, the paper has a number of limitations and short-comings which are in part difficult to fix and I'm therefore reluctant to recommend it for

publication. If the authors decide to submit a revised version, they should consider my concerns in detail and address them in their revisions.

My first comment to the measurements is that it is unfortunate that there are so few of them and that they were taken at 20° elevation angle. In my opinion, for flux measurements, a large number of zenith-sky observations taken with short integration times would be more useful as this reduces uncertainties in the integration of the flux and also in the light path. The three rounds around the airport have all relatively large gaps and I think that a statistically much more meaningful data set could be measured by the authors in a relatively short time when using one of their more modern instruments.

My second comment on the measurements is that the authors need to discuss the possibility of contamination of their measurements by car emissions on the highways they are driving on. The high NO2 signal close to the airport is suggesting that this NO2 was emitted from the airport but at least part of the NO2 could also be from road traffic, in particular as the instrument was integrating at 20° elevation along the road.

My third comment on the measurements is that the fundamental problem of estimating variable emissions using the flux approach needs to be discussed in more detail. As the authors will agree, the computation of emissions from measurements of the flux through an envelope around the source will only be straight forward if everything is in stationary state. If emissions change over time, they can only be estimated taking the distance between measurement and emission and wind speed and direction into account. This is then more a plume transect than a flux method, and for accurate results, plume dispersion and also NOx plume chemistry need to be considered. In the case of the airport this is further complicated by the presence of several moving sources at different distances from the measurement. While the manuscript tries to address parts of this issue, I think a much more detailed discussion is needed here.

My fourth comment on the measurements is that in the introduction, the problem of separation between aircraft and other emissions on the airport area is discussed in

some detail, suggesting that the current study will improve in this respect on previous ones. However, I do not see how the car-DOAS measurements can solve this problem. While integrating vertically through the atmosphere helps to include aircraft emissions during the first part of the ascent in the measurements, all ground-level emissions will still contribute to the emission estimate, independent of their source.

The main argument the authors make for their ability to estimate aircraft emissions during take-off is the correlation of NO2 flux with the number of aircraft take-offs. While I would expect to see such a correlation, the small number of three rounds in combination with the difficulty to quantify the number of take-offs and the other uncertainties makes this analysis pointless in my eyes. As stated above, I think that with limited effort, the authors could make this a much more meaningful study by adding a few more data points.

Although the manuscript is clearly written, it lacks some relevant information: The location, time and geometry of the background spectrum needs to be given, the method of computing the tropospheric columns should briefly be repeated and a figure showing the tropospheric vertical columns from the three rounds as a function of position along the track would be helpful to read the absolute numbers and to see their variability in time. Adding the wind arrow in Figures 3a-c would also be useful.

As a last comment I missed a discussion of the numbers the authors derive in the context of existing NOx emission estimates for aircrafts and airports. I'm sure that Frankfurt airport has to report NOx emissions to the environmental agencies and it would be appropriate to compare the values obtained from this study with those from the operators and also with numbers published in earlier work.